# Episiotomy for Medical Indications during Vaginal Birth—Retrospective Analysis of Risk Factors Determining the Performance of This Procedure

**DOI:** 10.3390/jcm11154334

**Published:** 2022-07-26

**Authors:** Grażyna Bączek, Sylwia Rychlewicz, Dorota Sys, Patryk Rzońca, Justyna Teliga-Czajkowska

**Affiliations:** 1Department of Obstetrics and Gynecology Didactics, Faculty of Health Sciences, Medical University of Warsaw, 00-575 Warsaw, Poland; justyna.teliga-czajkowska@wum.edu.pl; 2St. Sophia’s Specialist Hospital, Żelazna Medical Center, 01-004 Warsaw, Poland; s.rychlewicz@szpitalzelazna.pl; 3Department of Reproductive Health, Centre of Postgraduate Medical Education, 01-004 Warsaw, Poland; dorota.sys@cmkp.edu.pl; 4Department of Human Anatomy, Faculty of Health Sciences, Medical University of Warsaw, 5 Chałubińskiego St., 02-004 Warsaw, Poland; przonca@wum.edu.pl

**Keywords:** episiotomy, risk factors, vaginal birth

## Abstract

The WHO (World Health Organization) recommends that the percentage of perineal incisions should not exceed 10%, indicating that this is a good goal to achieve, despite the fact that it is still a frequently used medical intervention in Poland. The risk factors for perineal incision that have been analyzed so far in the literature allow, among others, to limit the frequency of performing this procedure. Are they still valid? Have there been new risk factors that we should take into account? We have conducted this study to find the risk factors for performing perineal incision that would reduce the frequency of this procedure. The aim of the study was to check whether the risk factors that were analyzed in the literature are still valid, to find new risk factors for perineal incisions and to compare them among Polish women. This was a single-center retrospective case-control study. The electronic patient records of Saint Sophia’s Hospital in Warsaw, Poland, a tertiary hospital was used to create an anonymous retrospective database of all deliveries from 2015 to 2020. The study included the analysis of two groups, the study group of patients who had had an episiotomy, and the control group-patients without an episiotomy in cases where an episiotomy was indicated. A logistic regression model was developed to assess the risk factors for perineal laceration. Independent risk factors for episiotomy in labor include oxytocin use in the second stage of labor (OR (Odds Ratio) = 6.00; 95% CI (Confidence Interval): 4.76–7.58), the supply of oxytocin in the first and the second stage of labor (OR = 3.18; 95% CI: 2.90–3.49), oxytocin use in the first stage of labor (OR = 2.72; 95% CI: 2.52–3.51), state after cesarean section (OR = 2.97; 95% CI: 2.52–3.51), epidural anesthesia use (OR = 1.77; 95% CI: 1.62–1.93), male gender (OR = 1.10; 95% CI: 1.02–1.19), and prolonged second stage of labor (OR = 1.01; 95% CI: 1.01–1.01). A protective factor against the use of an episiotomy was delivery in the Birth Centre (OR = 0.43; 95% CI: 0.37–0.51) and mulitpara (OR = 0.31; 95% CI: 0.27–0.35). To reduce the frequency of an episiotomy, it is necessary consider the risk factors of performing this procedure in everyday practice, e.g., limiting the use of oxytocin or promoting alternative places of delivery.

## 1. Introduction

In recent decades, the development of medicine in obstetrics has contributed to the improvement of the model of care for the mother and child in labor, but it also had consequences (e.g., epidural anesthesia helps a woman cope with labor pain, but also increases the frequency of perineal incisions) [1,2]. The use of medical interventions in life-threatening situations during childbirth (e.g., perineal incision in case of suspected fetal hypoxia during the second stage of labor) has a positive impact on the condition of the child, but the use of these procedures in situations that do not require it may be risky for the mother [3,4]. Medicalizing childbirth is defined as the use of unnecessary medical procedures that are applied without justification or indications in each case [5]. An example of such an intervention is, inter alia, episiotomy.

An episiotomy is a surgical incision to the vagina and perineum made by a qualified medical staff to widen the vaginal opening. The first documented perineal incision was made over 270 years ago. The frequency of this procedure increased significantly in the first half of the 20th century as women increasingly began to give birth in hospitals. There are studies which show that the presence of a midwife during labor correlates with a smaller number of perineal incisions, e.g., in the Birth Center, where only midwives work [6,7]. Another study found that the presence of medical residents poses the risk of an episiotomy [8]. In this regard, it has been noticed that the use of an episiotomy carries a risk for the mother. Complications that are associated with this procedure include bleeding, pain and discomfort around the perineum (including problems with sitting), scarring of wounds, dyspareunia, or complications in subsequent births [9]. One of the Polish studies from 2018 showed that the episiotomy procedure prevents third and fourth degree lacerations, however in critical conditions such as shoulder dystocia, instrumental deliveries, occiput-posterior position, fetal macrosomia, and NRFHR might increase the risk for third or fourth degree perineal tears [10,11].

The WHO does not recommend the use of a routine perineal incision in women who are experiencing spontaneous vaginal delivery, but recommends that the percentage of perineal incisions does not exceed 10%, indicating that it is a “good goal to achieve” [12,13].

In Poland, in 2005, perineal incisions were performed in 55.5% of all deliveries, and in 2013 in 42.7% [14,15]. According to the report of the Rodzić po Ludzku Foundation of 2021, 68% of respondents from 2015–2016 (according to the report of the Supreme Audit Office) declared that they had had an episiotomy, in 2018 (according to the Foundation’s report) the percentage was 55%, and in 2021 (according to the same report) it was 51% [16].

Scientific evidence confirms that an episiotomy should be used in specific situations where the benefits of the incision outweigh the possible complications of the procedure (routine use of this procedure is contraindicated). However, it has not been confirmed that an episiotomy is necessary under any circumstances [17].

Accordingly, factors that predispose a mother to the use of an episiotomy are important. Suggested indications in the literature included premature delivery, breech presentation, fetal macrosomia, shoulder dystocia, operative delivery, abnormal fetal heart function or an inelastic, and a tear-prone perineum. However, it was analyzed whether such situations really constitute indications for an episiotomy or not and it was concluded that research in this area should be continued [9,18].

Despite the WHO guidelines, perineal incision is still a commonly used medical intervention in Poland [11]. We have conducted this study to find the risk factors for performing perineal incision that would reduce the frequency of this procedure. The aim of the study was to check whether the risk factors that were analyzed in the literature are still valid, to find new risk factors for perineal incisions, and to compare them among Polish women giving birth in hospital with the third level of reference in cases where an episiotomy was indicated. Most of the literature in this area does not specify whether the perineal incision is systematic/routine or is performed due to medical indications. Updating the risk factors that are related to perineal incisions and searching for new ones taking into account its causes (resulting from medical indications), is a gap in the studied area, therefore, an attempt was made to fill it.

## 2. Materials and Methods

This was a single-center retrospective case-control study. The Strobe guidelines for case-control studies were used to ensure the proper reporting of results [19]. The study has received the approval from the Bioethics Committee of the Medical University of Warsaw (No. AKBE/204/2021). This was a retrospective anonymized data analysis; therefore, no individual patient consent was needed.

The electronic patient records of Saint Sophia’s Hospital in Warsaw, Poland, a tertiary hospital with the largest number of deliveries per year, were used to create an anonymous retrospective database of all the deliveries from 2015 to 2020. In this hospital, one can give birth in a standard delivery room or in the Birth Center (a midwife-led unit for natural births without medicalization—no epidural anesthesia, no oxytocin use). Healthy women with low-risk pregnancy and without any health burden can give birth in the Birth Center. Approximately 8% of vaginal births happen at the Birth Center every year. In the delivery room, both the first and the second stage of labor are supervised by the midwife and the doctor in close cooperation. In the Birth Center, only the midwife gives birth without the presence of a doctor. The number of births started at the Birth Center in the analyzed period was 3875 (100%), of which 10% (398) were transferred to the delivery room. In 71 patients (1.7%) cesarean section and in 25 women (0.9%) operative delivery were performed. This dataset was generated using electronic medical records that were collected by medical personnel. Therefore, there is no recall bias. Additionally, the dataset was cross-checked for inconsistencies and any that were detected were verified.

Multiple pregnancies and deliveries before 38 weeks of gestation were excluded from the analysis. Shoulder dystocia, which is an obvious indication for episiotomy, has also been excluded Our study is retrospective, not prospective, so we excluded this risk factor. Neonates with major birth defects or abnormal karyotype were also excluded [20,21]. The study included the analysis of two groups: the study group consisting of patients with an episiotomy that was performed and the control group-patients without an episiotomy being performed. The only criterion for inclusion in the study group was the performance of an episiotomy in cases where an episiotomy was indicated.

In the process of analyzing electronic documentation, the following information was obtained: age, place of residence, education, marital status, parity, gravidity, gestational diabetes, diabetes mellitus, pregnancy hypertension, pre-pregnancy hypertension, pregnancy cholestasis, VBAC, obesity, BMI, maternal smoking, place of childbirth, family childbirth, oxytocin in 1st stage, oxytocin in 2nd stage, oxytocin in 1st and 2nd stages, epidural anaesthesia, duration of 1st stage, duration of 2nd stage, sex of child, the fetus presentation, perineal laceration, degrees of perineal laceration, blood loss, duration of stay, birth weight, length, head and chest circumference, and the condition of the newborn (Apgar). Blood loss is measured visually and standard childbirth blood loss is assumed to be 400 mL. The documentation of 40,007 deliveries was analyzed, of which, based on the adopted criteria, 27,340 cases qualified for the analysis. However, due to the lack of data in the medical records, 19,599 cases qualified for the final analysis.

### Statistical Analysis

The data that were obtained in the documentation analysis process was subjected to statistical analysis, which was performed using the R language in the RStudio environment. Qualitative data are presented as numbers (*n*) and case percentages (%). Quantitative data were presented as the mean (M) and standard deviation (SD). The Pearson Chi-square test was used to assess the dependence within the qualitative variables. Quantitative variables were compared using the Student’s t-test with the assessment of homogeneity of variance with the Brown Forsythe test. A logistic regression model was developed to assess the risk factors for perineal laceration. The backward stepwise method was used in the construction. The model data are presented as odds ratios (OR) together with the 95% confidence interval (95% CI). The usefulness of the model was assessed using the ROC method with the determination of the cut-off point with the tangent method. The level of statistical significance was set at *p* < 0.05.

## 3. Results

The episiotomy was more common in younger women giving birth (30.62 vs. 31.32 years), city residents (88.2%), and the found correlations were statistically significant (*p <* 0.05). There were no significant correlations between the episiotomy and the marital status of the studied women (*p* > 0.05). Detailed data are presented in Table 1.

The episiotomy was more common in primiparous women who were examined in their first pregnancy, during their first delivery, in women who had been diagnosed with gestational diabetes, had a history of cesarean section, and with a higher BMI. Moreover, it has been shown that the episiotomy was performed more often in women who gave birth in the delivery room than in the Birth Center, who were treated with oxytocin in the first stage of labor, the second stage of labor, and during the first and the second stage of labor, and in women who were treated with epidural anesthesia. In the group of women who underwent an episiotomy, it was found that the duration of the first and the second stage of labor was longer compared to the women who did not undergo this procedure. It was also found that the episiotomy was performed more often in women who had a partner during labor (family childbirth) or gave birth to sons. The above results were statistically significant. (*p* < 0.05). The results are presented in Table 2.

Analyzing the perinatal results of the mother following the episiotomy, it was found that perineal lacerations were more common in the control group, i.e., in women who had not had the episiotomy. It was noticed that the loss of blood volume during labor was greater in the group of patients who underwent an episiotomy. A longer hospitalization was found more often in women who underwent perineal incision. Among women giving birth with episiotomy, it was noticed that the birth weight of the child and the circumference of the head were greater, and the condition of the newborn after birth was more often assessed at ≤ 7 points on the Apgar scale in 1′ and 3′ minutes The above results were statistically significant (*p* < 0.05). These results are presented in Table 3.

Logistic regression models were developed to identify the risk factors for an episiotomy. The univariate analysis showed a significant increase in the risk of an episiotomy in the case of oxytocin use in the second stage of labor (OR = 9.23; 95% CI: 7.41–11.49), oxytocin use in the first and the second stage of labor (OR = 4.59; 95% CI: 4.23–4.98), and the use of epidural anesthesia (OR = 3.58; 95% CI: 3.32–3.85). The same correlation was found in primiparous women (OR = 3.43; 95% CI: 3.18–3.70) and in cases after cesarean section (OR = 2.05; 95% CI: 1.80–2.34). A significant protective factor against the use of an episiotomy was delivery in the Birth Center (OR = 0.18; 95% CI: 0.15–0.21). The remaining risk factors are shown in Table 4.

Multivariate analysis showed that the independent risk factors for episiotomy in labor were: the first delivery, post-cesarean section cases, epidural anesthesia use, prolonged second stage of delivery, male gender, and a greater birth weight of the child (Table 5). Importantly, a correlation was observed between being multiparous and the duration of the second stage of labor. After including it in the analysis, it was observed that the prolonged second stage of labor in a multiparous woman increased the risk of incision, a unit change in the duration of the second stage of labor (by 1 min) increased the chance of incision by 1.01 times. Being a multiparous woman itself, in turn, proved to have a lower chance of making the incision. It should be noted that the administration of oxytocin in the second stage of labor increases the risk of an episiotomy up to six times (OR = 6.00; 95% CI: 4.76–7.58), and its administration in the first and the second stage of labor more than three times (OR = 3.18; 95% CI: 2.90–3.49). An independent protective factor against the occurrence of an episiotomy was birth in the Birth Center (OR = 0.43; 95% CI: 0.37–0.51). The predictive value of the model has a sensitivity of 71% and a specificity of 70% (AUC = 0.783).

## 4. Discussion

The WHO recommends that the percentage of perineal incisions does not exceed 10%, indicating that it is a ‘good goal to achieve’ [12,13]. Despite this, in Poland it is still a frequently performed medical intervention [11]. We have conducted this study to find the risk factors for performing perineal incision that would reduce the frequency of this procedure. The aim of the study was to check whether the risk factors that have been analyzed so far in the literature are still valid, to find new ones, and to compare them with the results that were obtained among Polish women. Most of the literature in this area does not specify whether the perineal incision is systematic/routine or is performed due to medical indications. Updating risk factors that are related to perineal incisions and searching for new ones taking into account its causes (resulting from medical indications), is a gap in the studied area, therefore, an attempt was made to fill it. The conducted study indicated that independent risk factors for an episiotomy in labor were the first delivery, prolonged second stage of labor, state after cesarean section, epidural anesthesia, male gender, and a higher birth weight. An important protective factor against the use of an episiotomy was the birth in the Birth Center. In addition, a significant increase in the risk of an episiotomy was observed in the case of the use of oxytocin in the second stage of labor, oxytocin administration in the first and the second stage of labor, epidural anesthesia in the cases of primiparous women, and the state after the cesarean section.

Based on the results of our own research, it was found that one of the risk factors for an episiotomy is the first labor. Soleimanzadeh et al. (2020) found that 90% of Iranian women who had experienced an episiotomy had their first delivery, while for women who did not have an episiotomy, only 4% [22]. The results of the research by Beyene et al. (2020), Rasouli et al. (2016), and Aguiar B. et al. (2020) also confirm a significant relationship between the first delivery and more frequent performance of episiotomy [8,23,24]. A study by Wu L. at al. (2013) assessed the risk factors and causes of an episiotomy that were reported by midwives. The most common midwife-reported reason for episiotomy among primiparous women was primiparity, and among multiparous women was fetal distress and poor maternal effort. [25]

Beyene et al. (2020) and Pebolo et al. (2020) found that a prolonged second stage of labor was a factor that was significantly associated with a higher risk of an episiotomy [8,26]. The same statement can be found in a systematic review by Clesse at al. (2020) [2]. This assumption was also confirmed in our study. A study by Tefer et al. (2019) showed that the duration of the second stage of labor that was over 90 min was an independent risk factor for an episiotomy [27].

The results of our own research have shown that the history of a single cesarean section significantly increases the risk of an episiotomy. In the already cited study by Clesse et al. (2019), they also noted that the identification of an earlier cesarean section in the mother poses a risk of an episiotomy [28] Charitou et al. (2019) and Carvalho et al. (2010) showed that an episiotomy was more common in cases with VBAC [29,30]. On the other hand, Braga et al. (2014) found no correlation between an episiotomy and a history of cesarean section [7].

According to our study, it was noticed that the use of epidural anesthesia significantly increases the risk of an episiotomy. Ballesteros-Meseguer et al. (2016) and Shmueli et al. (2017) also showed a tendency to perform an episiotomy in women who received epidural anesthesia [31,32]. In contrast, in a study by Liu et al. (2021), no significant correlations were found between epidural anesthesia and the frequency of episiotomy [33].

The male gender was found to be another independent risk factor of an episiotomy in our study. However, in a Spanish study by Hernandez et al. (2014) [34] and the study from Ethiopia by Marai (2002) [35], no difference between the sex of the child and the episiotomy procedure was shown.

Our study showed that a higher birth weight of the newborn significantly increases the risk of an episiotomy. Beyene et al. (2020) found that a birth weight ≥4000 g increases the risk of an episiotomy [8]. Ballesteros-Meseguer et al. (2016) noticed a tendency in their study not to perform an episiotomy in newborns weighing <2500 g, while in those with a weight of 2500–4000 g and >4000 g, the tendency is to do so [31]. Other researchers have also shown a relationship between the birth weight >4000 g and an increased frequency of an episiotomy [2,27,31,36,37]

Rasouli et al. (2016) found a significant correlation between the use of oxytocin in labor and an episiotomy [23]. Other researchers have also come to similar conclusions [8,31,38]. The study by Woretaw et al. (2021) showed that women who had used oxytocin during childbirth were 2.73 times more likely to have a perineal incision than those who had not used oxytocin. This may be due to the fact that the incorrect use of oxytocin causes too strong contractions of the uterus, which may affect the child’s heartbeat [38]. The studies by Karahan et al. (2018), who emphasized that the administration of oxytocin seems to interfere with the coordination of contractions of the uterine and pelvic floor muscles [39]. This may result in a prolongation of the second stage of labor and consequently an increased risk of incision [Beyene et al. (2020) and Pebolo et al. (2020)], which was also shown by the results of own research [8,26]. In our study, an increase in the risk of an episiotomy was observed in the case of the use of oxytocin in the second stage of labor and the administration of oxytocin in the first and second stage of labor.

In addition, Espada-Trespalacios et al. (2021) also noted in their studies the high rate of perineal incision that was observed in the examined women who received oxytocin. According to the researchers, this can be explained by a higher frequency of operative deliveries [40]. In turn, Desplanches et al. (2022) found that mediolateral episiotomy reduces the risk of obstetric anal sphincter injuries among nulliparous women that are undergoing operative vaginal delivery [41].

A protective factor against the occurrence of an episiotomy in our study was a delivery in the Birth Center, where only midwives provide delivery. Słoma et al. (2020) presented that medical procedures, including the episiotomy procedure, were much more frequent among patients from hospitals and private clinics than from the Birth Center [6].

The Birth Center is a place where only healthy women with low-risk pregnancy and without any health problems can give birth. The experience of childbirth is not disturbed by the use of unexplained medical procedures. Laboring women stay under the care of a midwife, who restricts interventions to the essential safe minimum, such as episiotomy [42]. Suto et al. (2015) found a lower percentage of perineal incisions during spontaneous, physiological deliveries in low-risk women at the Birth Center in Japan. [43] There have been reports that the rates of perineal incision are lower in the Birth Center, but there was no significant difference in the incidence of third- and fourth-degree rupture between the groups in the studies that analyzed this measure [44,45,46]. A study by Cromi et al. (2015) draws attention to the role of midwives. And so, in the studied hospital, the overall percentage of episiotomy in primiparous women was 40.6%, and the percentage of perineal incision that was performed by a midwife ranged from 5.6% to 73.9% (*p* < 0.0001). Logistic regression indicated that the individual midwives were all associated with episiotomy use [47]. Whereas Beyene et al., (2020) noted that factors that remained significantly associated with a higher risk of an episiotomy included delivery with medical residents. The authors of this study showed that births with medical residents were three times more likely to be performed with an episiotomy than those in which midwives participated. This phenomenon is explained by the practices of the center where the study was conducted—residents were invited to manage the birth in two cases. The first one is complicated births that are carried out by senior health professionals in which episiotomy is highly required. The second reason is childbirth, which required the presence of a doctor. Perhaps leaving residents unattended by senior doctors also contributes to an increase in the percentage of perineal incisions [8]. In addition, a study by Howden et al. (2004) found that the predictor of an episiotomy was the type of physician, namely women attending private practitioners had a seven-fold increased risk of an episiotomy [48]. In addition, a study by Zhang M. et al. (2018), in West China, among women >28 weeks of pregnancy, it was noticed that midwives with a bachelor’s degree and obstetricians with a doctor’s degree performed an episiotomy the most frequently [49].

In our study, other perinatal results were analyzed, which turned out to be statistically significant (age, place of residence, gestational diabetes, high BMI, increased blood volume loss, perineal rupture, and a lower Apgar score in 1’ and 3’ minutes of life).

In our study, the episiotomy was significantly more often performed in the group of younger women that were giving birth and city residents. Soleimanzadeh et al. (2020) proved that the mean age of mothers who had an episiotomy was significantly lower than that of mothers without an episiotomy [22]. Aguiar B. et al. (2020) came to the same conclusion [24]. Rasouli et al. (2016) did not show any correlation between this medical intervention and the place of residence, profession, or race, but found a correlation between the episiotomy and the mother’s education (more often among women with higher education) [23]. Our study found no correlation between the episiotomy and the education and marital status of women. On the other hand, Fikadu et al. (2020) observed that unmarried women often had an episiotomy significantly more [50].

In our own research, it was observed that the episiotomy was significantly more frequent in the group of women that were diagnosed with gestational diabetes and high BMI. Khajehei et al. (2020) found that the presence of diabetes did not influence the relative risk of an episiotomy [51]. Women who develop gestational diabetes are more likely to have stillborn or to deliver macrosomic newborns, so they may require surgical intervention such as an episiotomy or cesarean section [52]. On the other hand, the correlation between a low BMI and a greater number of episiotomies was observed by Fikadu et al. (2020). In Ethiopia, mothers with a BMI < 25 kg/m^2^ were almost three times more likely to have an episiotomy during labor than mothers with a BMI ≥ 25 kg/m^2^ [50].

Our study showed that increased blood loss was greater in women who had an episiotomy. In a study by Amorim et al. (2017), no differences were found between episiotomy and blood loss in childbirth [17]. This is in contrast to the study by Girault et al. (2018), where it has been shown that the risk factor for increased blood loss after childbirth is an episiotomy [53].

In our study, it was noticed that perineal lacerations were significantly more common in women who had no episiotomy performed. In the same group, a significantly higher percentage of first degree perineal lacerations was observed. Second and third degree perineal lacerations were more common among women who underwent an episiotomy. Gebuza et al. (2018) found that reducing the frequency of episiotomies increased the incidence of first and second degree perineal lacerations. Their study showed that an episiotomy protects women from third and fourth degree perineal lacerations [11]. Woretaw et al. (2021) noted a significant correlation between a lower percentage of perineal lacerations while increasing the percentage of episiotomies [38]. Moreover, Steiner et al. (2012) proved that the perineal incision in critical situations, such as shoulder dystocia or vaginal delivery, is an independent risk factor for Grade III or IV perineal tear [10]

In our study, among those who underwent an episiotomy, a greater circumference of the head of the newborn was noticed and that the condition of the newborn after birth was more often assessed at ≤7 points on the Apgar scale in the 1’ and 3’ minutes. Rasouli et al. (2016) made similar observations and found a significant correlation between the episiotomy and a lower Apgar score in the first minute of life and a larger head circumference of the newborn [23]. Soleimanzadeh et al. (2020) noticed that in the group of women who underwent an episiotomy, the newborn’s score according to the Apgar scale in the first and the fifth minutes of life was significantly lower [22].

The use of an episiotomy is one of the most common procedures that is performed during labor and should be restricted [11]. The knowledge about the risk factors that are associated with an episiotomy can help reduce and prevent this medical intervention especially in Poland, where the percentage of perineal incisions is still higher than as in WHO recommendations [11,13,54].

We analyzed a large group and the studies were conducted in a place that is known for high standards of perinatal care and many years of clinical experience, but there were some limitations. One of them being the study was carried out in one center where the internal standards of the center may differ from other hospitals, which may affect the results that were obtained. Another limitation is that the analyzed data came from electronic documentation, which resulted in the restriction of access to certain variables. Therefore, the next step should be to extend the research to other clinical centers and to add other variables to the analysis.

## 5. Conclusions

The univariate analysis showed a significant increase in the risk of an episiotomy in cases of oxytocin use during labor, epidural anesthesia, in primiparous women, and those in labor after previous cesarean section. The protective factor turned out to be childbirth in the Birth Center. The independent risk factors for episiotomy in labor include the first delivery, the state after cesarean section, the use of epidural anesthesia, prolonged second stage of labor, male gender, and a higher birth weight of the child.

To reduce the frequency of episiotomies, it is necessary to take into account the risk factors of performing this procedure in everyday practice, e.g., limiting the use of oxytocin or promoting alternative places of delivery.

Evidence-based preconception health education can help reduce unnecessary medical interventions and thus protect women’s reproductive health.

It seems appropriate to organize systematic training in the field of delivery management and indications for perineal incisions that are conducted by midwives that are experienced in working with the parturient women.

## Figures and Tables

**Table 1 jcm-11-04334-t001:** Characteristics of the population.

	Study Group Episiotomy N = 4001 *n* = (%)	Control Group No Episiotomy N = 15,598 *n* = (%)	ALL *n* = (%)	*p*-Value
Age [years]				<0.0001
<18	7 (0.17)	17 (0.11)	24 (0.12)	
18–25	411 (10.27)	1248 (8.00)	1659 (8.47)	
26–30	1623 (40.56)	5435 (34.84)	7058 (36.01)	
31–35	1481 (37.02)	6417 (41.14)	7898 (40.30)	
>35	479 (11.97)	2481 (15.91)	2960 (15.10)	
Place of residence				<0.0001
City	3516 (88.2)	13,409 (86.2)	16,925 (86.6)	
Village	470 (11.8)	2151 (13.8)	2621 (13.4)	
Education				
Higher education	2769 (88)	12,810 (87.4)	15,579 (87.5)	0.08
Secondary education	362 (11.5)	1727 (11.8)	2089 (11.7)	
Primary education	14 (0.4)	120 (0.8)	134 (0.8)	
Marital status				0.07
In a relationship	3021 (80.5)	12,455 (81.8)	15,476 (81.6)	
Single	730 (19.5)	2765 (18.2)	3495 (18.4)	

N—group size; *n*—number of observations.

**Table 2 jcm-11-04334-t002:** Analysis of the incidence of perineal incisions among the selected variables occurring before delivery.

	Study Group Episiotomy N = 4001 *n* = (%) * M ± SD	Control Group No Episiotomy N = 15,598 *n* = (%) * M ± SD	ALL *n* = (%) * M ± SD	*p*-Value
Parity				
Primipara	2859 (71.5%)	6586 (42.2%)	9445 (48.2)	<0.0001
Multipara	1142 (28.5%)	9012 (57.8)	10,154 (51.8)	
Gravidity				
1	2462 (61.5)	5637 (36.1)	8099 (41.3)	
2	1047 (26.2)	5805 (37.2)	6852 (35.0)	
3	351 (8.8)	2569 (16.5)	2920 (14.9)	<0.0001
4	94 (2.3)	948 (6.1)	1042 (5.3)	
5 and more	47 (1.2)	639 (4.1)	427 (2.2)	
Gravidity	1.56 ± 0.85 *	2.07 ± 1.17 *	1.97 ± 1.13 *	<0.0001
Parity				
1	2859 (71.5)	6586 (42.2)	9445 (48.2)	
2	947 (23.7)	6386 (40.9)	7333 (37.4)	
3	163 (4.1)	1946 (12.5)	2109 (10.8)	<0.0001
4	27 (0.7)	454 (2.9)	481 (2.5)	
5 and more	5 (0.1)	226 (1.4)	156 (0.8)	
Parity	1.34 ± 0.61 *	1.81 ± 0.91 *	1.72 ± 0.88 *	<0.0001
Gestational diabetes				0.01
No	3581 (89.5)	14,177 (90.9)	(17,758 (90.6)	
Yes	420 (10.5)	1421 (9.1)	1841 (9.4)	
Diabetes mellitus				0.18
No	3994 (99.8)	15,583 (99.9)	19,577 (99.9)	
Yes	7 (0.2)	15 (0.1)	22 (0.1)	
Pregnancy hypertension				0.16
No	3889 (97.2)	15,222 (97.6)	19,111 (97.5)	
Yes	112 (2.8)	376 (2.4)	488 (2.5)	
Pre-Pregnancy hypertension				0.8
No	3979 (99.5)	15,507 (99.4)	19,486 (99.4)	
Yes	22 (0.5)	91 (0.6)	113 (0.6)	
Pregnancy cholestasis				0.79
No	3958 (98.9)	15,438 (99)	19,396 (99)	
Yes	43 (1.1)	160 (1)	203 (1)	
VBAC				<0.0001
No	3642 (91)	14,882 (95.4)	18,524 (94.5)	
Yes	359 (9)	716 (4.6)	1075 (5.5)	
Obesity				0.06
No	3886 (97.1)	15,231 (97.6)	19,117 (97.5)	
Yes	115 (2.9)	367 (2.4)	482 (2.5)	
BMI	22.2 ± 3.45 *	22 ± 3.44 *	22.04 ± 3.45 *	0.03
Maternal smoking				0.55
No	3981 (99.5)	15,531 (99.6)	19,512 (99.6)	
Yes	20 (0.5)	67 (0.4)	87 (0.4)	
Place of childbirth				<0.0001
Delivery room	3814 (95.3)	12,229 (78.4)	16,043 (81.9)	
Birth Center	187 (4.7)	3369 (21.6)	3556 (18.1)	
Family Childbirth				<0.0001
No	1015 (25.4)	9536 (61.1)	10,551 (53.8)	
Yes	2986 (74.6)	6062 (38.9)	9048 (46.2)	
Oxytocin in 1st stage				<0.0001
No	3950 (98.7)	15,504 (99.4)	19,454 (99.3)	
Yes	51 (1.3)	94 (0.6)	145 (0.7)	
Oxytocin in 2nd stage				<0.0001
No	3736 (93.4)	15,479 (99.2)	19,215 (98)	
Yes	265 (6.6)	119 (0.8)	384 (2.0)	
Oxytocin in 1st and 2nd stages				<0.0001
No	2511 (62.8)	13,812 (88.5)	16,323 (83.3)	
Yes	1490 (37.2)	1786 (11.5)	3276 (16.7)	
EA				
No	2143 (53.6)	12,555 (80.5)	14,698 (75)	<0.0001
Yes	1858 (46.4)	3043 (19.5)	4901 (25)	
Duration of 1st stage [min]	346.65 ± 156.12 *	291.91 ± 143.66 *	303.09 ± 147.94 *	<0.0001
Duration of 2nd stage [min]	37.98 ± 23.35 *	25.35 ± 20.55 *	27.93 ± 21.75 *	<0.0001
Sex of Child				0.01
Female	1932 (48.3)	7898 (50.6)	9830 (50.2)	
Male	2069 (51.7)	7700 (49.4)	9769 (49.8)	
The fetus presentation				0.98
Breech	1 (0.0)	4 (0.0)	5 (0)	
Cephalic	4000 (100)	15,594 (100)	19,594 (100)	

* M ± Standard Deviation; SD—standard deviation; BMI—body mass index; VBAC—vaginal birth after cesarean delivery; EA—epidural analgesia.

**Table 3 jcm-11-04334-t003:** Analysis of the incidence of perineal incisions among the selected variables occurring after delivery.

	Study Group Episiotomy N = 4001 *n* = (%) * M ± SD	Control Group No Episiotomy N = 15,598 *n* = (%) * M ± SD	ALL *n* = (%) * M ± SD	*p*-Value
Perineal laceration				<0.0001
No	3301 (82.5)	11,998 (76.9)	15,299 (78.1)	
Yes	700 (17.5)	3600 (23.1)	4300 (21.9)	
Degrees of perineal laceration				<0.0001
None	3608 (90.2)	11,958 (76.7)	15,566 (79.4)	
1	323 (8.1)	3463 (22.2)	3786 (19.3)	
2	59 (1.5)	166 (1.1)	225 (1.1)	
3	10 (0.2)	8 (0.1)	18 (0.1)	
4	1 (0.0)	3 (0.0)	4 (0)	
Blood Loss	422.39 ± 160.81 *	371.63 ± 109.91 *	388.41 ± 131.17 *	<0.0001
Duration of stay [days]	4.43 ± 2.61 *	3.89 ± 3.07 *	4.0 ± 2.99 *	<0.0001
Birth Weight	3486.99 ± 428.04 *	3467.26 ± 414.23 *	3471.29 ± 417.15 *	0.01
Length	54.67 ± 2.59 *	54.58 ± 2.64 *	54.6 ± 2.63 *	0.05
Head circumference	34.75 ± 1.67 *	34.68 ± 1.71 *	34.7 ± 1.7 *	0.04
Chest circumference	33.99 ± 1.69 *	34.03 ± 1.99 *	34.03 ± 1.94 *	0.24
Apgar 1′				<0.0001
≤7	105 (0.8)	179 (0.5)	284 (0.6)	
>7	3896 (99.2)	15,417 (99.5)	19,313 (99.4)	
Apgar 3′				<0.0001
≤7	42 (1.1)	78 (0.5)	120 (0.6)	
>7	3946 (98.9)	15,497 (99.5)	19,443 (99.4)	
Apgar 5′				0.09
≤7	13 (0.3)	29 (0.2)	42 (0.2)	
>7	3986 (99.7)	15,568 (99.8)	19,554 (99.8)	
Apgar 10′				0.54
≤7	4 (0.1)	11 (0.1)	15 (0.1)	
>7	3991 (99.9)	15,577 (99.9)	19,568 (99.9)	

* M ± Standard Deviation; SD—standard deviation.

**Table 4 jcm-11-04334-t004:** Logistic regression for performing the episiotomy.

Effect		*p*-Value	Odds Ratio (OR)	Confidence Interval OR -95%	Confidence Interval OR 95%
Total fertility rate	Primipara	<0.0001	3.43	3.18	3.69
	Multipara		1.00		
Weight		0.01	1.00	1.00	1.00
Head circumference		0.04	1.02	1.00	1.04
Sex	Male	0.01	1.10	1.02	1.18
	Female		1.00		
Gestational diabetes		0.01	1.17	1.04	1.31

Obesity		0.06	1.23	0.99	1.52

Vaginal birth after cesarean delivery		<0.0001	2.05	1.796	2.34

Place	Birth Center	<0.0001	0.18	0.15	0.21
	Delivery room		1.00		
Duration of 1st period		<0.0001	1.00	1.00	1.00
Duration of 2nd period		<0.0001	1.02	1.02	1.03
Oxytocin in 1st stage		<0.0001	2.13	1.51	3.00

Oxytocin in 2nd stage		<0.0001	9.23	7.41	11.49

Oxytocin in 1st and 2nd stages		<0.0001	4.59	4.23	4.98

Epidural analgesia		<0.0001	3.58	3.32	3.85


**Table 5 jcm-11-04334-t005:** Multivariate regression for episiotomy.

Effect	AUC = 0.783; Sensitivity 71%, Specificity 70%			
	*p*-Value	Odds Ratio	Confidence Interval OR -95%	Confidence Interval OR 95%
Absolute term	0.00	0.08	0.06	0.11
Multipara	0.00	0.31	0.27	0.35
Male Gender	0.02	1.10	1.02	1.19
State after a cesarean section	0.00	2.97	2.52	3.51
Childbirth in the Birth Center	0.00	0.43	0.37	0.51
Oxytocin in 1st stage	0.00	2.72	1.89	3.92
Oxytocin in 2nd stage	0.00	6.00	4.76	7.58
Oxytocin in 1st and 2nd stages	0.00	3.18	2.90	3.49
Epidural analgesia	0.00	1.77	1.62	1.93
Newborn’s weight	0.00	1.00	1.00	1.00
Duration of 2nd period	0.00	1.01	1.01	1.01
Primiparous/Multiparous/Duration of 2nd period	0.00	1.01	1.01	1.01

## Data Availability

The data presented in this study are available on request from the corresponding author.

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
