# Peer review of "Episiotomy for Medical Indications during Vaginal Birth—Retrospective Analysis of Risk Factors Determining the Performance of This Procedure"

_jcm, 2022, doi:10.3390/jcm11154334_

Round 1

Reviewer 1 Report

This study is performed on a large number of cases (otherwise it would not have been relevant). The subject itself is of great interest, given the tendency to return to more home birth care. The authors point out very well the conditions that increase the incidence of episiotomy, but do not sufficiently emphasize some aspects such as oxytocic infusion as a risk factor. The lack of supervision of resident doctors is another aspect that is not sufficiently emphasized. Perhaps the study would gain more prestige if there were such emphasis. One limitation, otherwise noted by the authors, is that the study was conducted in only one medical center, knowing that there are differences between medical schools in different medical centers.

The study is worth. But for its improvement, I emphasize the need to emphasize the mentioned aspects

Author Response

Dear Reviewer,
Thank you very much for reviewing our article. Reply to comments is attached.
Kind regards
Corresponding author

Reviewer 2 Report

This manuscript is informative and reminds the indications of Episiotomy. It is not a rutin procedure and more or less, we are concerned about a surgical trauma on perineum.The manuscript is educative and reminds some facts about the most commonly performed minor surgical intervention

Author Response

Dear Reviewer,
Thank you very much for reviewing our article. Reply to comments is attached.
Kind regards
Corresponding Author

Reviewer 3 Report

jcm-1766360

I was asked to peer review this manuscript about episiotomy “ Episiotomy in the course of labor - retrospective analysis of risk factors determining the performance of this procedure”. It is a single-centre retrospective analysis based on 19 599 vaginal deliveries at term. The aim of the presented research was to search for risk factors of the necessity to perform an episiotomy among women giving birth in the hospital with the third level of reference. In addition, the perinatal outcomes were analyzed in study and control groups (Lines 75-76) .

  • First of all, I suggest the authors carefully read the instructions for the authors and also the aim of the journal: “ Articles: Original research manuscripts. The journal considers all original research manuscripts provided that the work reports scientifically sound experiments and provides a substantial amount of new information. Authors should not unnecessarily divide their work into several related manuscripts, although short Communications of preliminary, but significant, results will be considered. The quality and impact of the study will be considered during peer review.” Also the IMRAD structure and content .
  • Secondly, the manuscript needs good English proofreading

The title : Episiotomy in the course of labor ? means systematic episiotomy or episiotomy when indicated ?

The abstract must be revised : I suggest to have a question to answer and to follow a simple flow of your description. Also the structure IMRAD .

Keywords : vaginal birth instead of childbirth

Introduction : Lines 36-38: the development of medicine in obstetrics? means the medicalization of vaginal birth ?  

Also Lines 38-39 : medical intervention in life-threatening situations?

I suggest to focus on episiotomy , to the recommendations of WHO, to different percentages of episiotomies, and to have a gap in this topic to demonstrate . I suggest also references about episiotomies in countries with similarities with Poland , why discussing about residents in Ethiopia (ref 6 ) -lines 50-51.

Lines 66-67 : The published guidelines of the World Health Organization state that the role of an episiotomy in vaginal delivery is still 67 to be established [13]. - Is operative vaginal deliveries

Materials and methods :

First of all , you need to explain your sample size and confidence interval, margin of error etc

I recommend to figure the presentation of your sample population selection.

I suggest to explain your analyzed variables.

I don’t understand why shoulder dystocia was excluded? Episiotomy is performed before the delivery of the head of the baby, shoulder dystocia appears after that and this analysis is on the episiotomy.  (line 98) .

Results : you discuss about young women ? but you need categories of age for the demographic analysis . In table 1 is the median age of the subjects.

Table 2 with high-risk pregnancy is not very clear, it is the indication or the consequence for episiotomy ? And Oxytocin ?

How many patients gave birth assisted by midwives in the Birth Center, and from this sample all the patients gave birth by the vaginal way ? You may figure the samples, the proportion of cesarean section, the proportion of instrumental deliveries etc .

Lines 149-150 : did they longer hospitalization because of episiotomy?

Discussion : focus on episiotomy when indicated? What this study add ? and what are the future direction, or the future recommendations? Discussions are very confusing because we don’t understand if it is about the systematic use of episiotomy or episiotomy for a medical indication?

Conclusions: it is very important what is the proportion of episiotomies in this study ? and reported by the proportion of episiotomies in Poland , in the world and recommended by WHO ? And you may recommend a protocol or training for a vaginal birth or changing the mindset for practicing actual obstetrics. 

Author Response

(The authors gave the same response as above.)

Round 2

Reviewer 3 Report

No comments

This manuscript is a resubmission of an earlier submission. The following is a list of the peer review reports and author responses from that submission.

Round 1

Reviewer 1 Report

Episiotomy in the course of labor - retrospective analysis of risk factors determining the performance of this procedure

Thank you as you invited me to review this manuscript. This is an interesting study as it shows that any unnecessary intervention in the process of childbirth may put the mother at the greater risk. Despite its merits, this manuscript has some issues that should be resolved by authors. Please see my comments as follows:

Abstract:

  1. Please write about tools that you used to gather data and also statistical methods.

Introduction

  1. Authors stated that episiotomy is a surgical incision that made in vagina by a qualified obstetrician. While around the world, midwives can perform this procedure, why authors only mentioned obstetricians?
  2. One of the predisposing factors to perform episiotomy is the birth attendant (midwife or obstetrician). Obstetricians are more likely to perform episiotomy. Please add some information about the effect of birth attendant on rate of episiotomy.

Methods

  1. In my opinion the design of this study is more likely to be a comparative cross-sectional study and not a case-control study.
  2. Authors stated that medical records used from hospital or Birth House. Is records of Birth house merged in hospital records?
  3. Participation in preparedness educational classes of women is an important factor that missed by authors. Usually, women who participated in educational classes need less episiotomy.
  4. Please explain how blood loss measured in the normal vaginal delivery?
  5. Please provide some more information about how many percentages of birth happened in the hospital and how many in “Birth House”?
  6. Also provide some information about who (midwife or obstetrician) provide care for women during their first and second stages of labor

Results

  1. Please mention the total numbers of participants in the top cell of each table.
  2. Table 1: Please report p value=0.000, as p<0.0001
  3. Page 3, line 128: please clarify the meaning of the following sentence: “had a family childbirth”
  4. Page 6, when reporting APGAR score is according to first and five minutes after birth, why authors reported it as first and three minutes after birth?

Discussion

  1. Please write the aim of the study at the beginning of this section.

Author Response

Review 1

Dear Reviewer

Thank you very much for your thorough analysis of our article and for all valuable comments and tips.

The indicated remarks have been made in the manuscript. Details below:

Abstract:

  1. Please write about tools that you used to gather data and also statistical methods.

Information was added into abstract

Introduction

  1. Authors stated that episiotomy is a surgical incision that made in vagina by a qualified obstetrician. While around the world, midwives can perform this procedure, why authors only mentioned obstetricians?

Thank you for your attention, there was a translation mistake. We have already covered this mistake in the article.

  1. One of the predisposing factors to perform episiotomy is the birth attendant (midwife or obstetrician). Obstetricians are more likely to perform episiotomy. Please add some information about the effect of birth attendant on rate of episiotomy.

Thanks for your suggestion. We expanded the introduction to include this information.

Methods

  1. In my opinion the design of this study is more likely to be a comparative cross-sectional study and not a case-control study.

“A comparative cross-sectional study design is an approach whereby you compare an outcome among two population groups, collecting data at one point in time.”

“In case-control studies, participants are recruited on the basis of disease status. Thus, some of participants have the outcome of interest (referred to as cases), whereas others do not have the outcome of interest (referred to as controls). The investigator then assesses the exposure in both these groups.”

Based on the above definitions, we propose to keep referring to our study as a case-control study, because our study includes a single population and aims to assess the exposure in a study and control groups.

  1. Authors stated that medical records used from hospital or Birth House. Is records of Birth house merged in hospital records?

Yes, we have one electronic medical records system in the whole Medical Center.

  1. Participation in preparedness educational classes of women is an important factor that missed by authors. Usually, women who participated in educational classes need less episiotomy.

We agree that this is important factor, but we haven’t this information in our medical records.

  1. Please explain how blood loss measured in the normal vaginal delivery?

Blood loss is measured visually, standard childbirth blood loss is assumed to be 400 ml - information added in the text of the work.

  1. Please provide some more information about how many percentages of birth happened in the hospital and how many in “Birth House”?

We have added this sentence into material and methods section

  1. Also provide some information about who (midwife or obstetrician) provide care for women during their first and second stages of labor

In the delivery room, both the first and the second stage of labor are supervised by the midwife and the doctor in close cooperation. Only in the Birth House, only the midwife gives birth without the presence of a doctor (information added in the text of the work)

Results

  1. Please mention the total numbers of participants in the top cell of each table.
  2. Table 1: Please report p value=0.000, as p<0.0001
  3. Page 3, line 128: please clarify the meaning of the following sentence: “had a family childbirth”
  4. Page 6, when reporting APGAR score is according to first and five minutes after birth, why authors reported it as first and three minutes after birth?

Thank you for your valuable comments. Points 1 and 2 have been corrected.

Point 3. Had a family childbirth means that women had a partner during labour (family childbirth). The explanation is included in the article.

Point 4

Thank you for paying attention to this aspect of the newborn's Apgar score. The assessment of the newborn on the Apgar scale in the center where the documentation was analyzed is performed by the medical team at the 1st, 3rd, 5th and 10th minutes, which was included in the results of our research. Additionally, in the description of the results, the assessment at the 3rd minute was taken into account due to the statistically significant relationship between the episiotomy and the Apgar score at the 3rd minute after delivery in the newborn..   

Discussion

  1. Please write the aim of the study at the beginning of this section.

Thanks for your suggestion. The aim was added to the discussion.

Thanks again for all comments,

I hope that all the corrections made will be accepted.

Reviewer 2 Report

Dear authors,

Thank you for submitting the script on the topic of episiotomy for me to review.

The topic itself is well known and the subject of many controversial discussions.  

The strength of your work is the high number of births included in the analysis. However, I still see room for improvement in the discussion and interpretation. The linguistic quality of the work is not objectionable. 

So far, I do not understand how your study provides significant new information. Shmueli et al reported in 2017 in Shmueli A, Gabbay Benziv R, Hiersch L. et al. Episiotomy - risk factors and outcomes. J Matern Fetal Neonatal Med 2017; 30: 251-276 doi:10.3109/14767058.2016.1169527 on risk factors. 
New additions were male sex and use of oxytocin. 

Particularly with oxytocin as an influencing factor, I miss the discussion of causation. Does oxytocin also affect pelvic floor muscle tone, causing a delay in the child's egress?

I find the reference to a lower risk of episotomy in a birth center problematic without comment. The birth center is run by midwives who are fundamentally very critical of intervention in terms of episiotomy. Similarly, one could argue that midwives rarely perform apendectomies. 
Here are two articles from a separate review on the topic: 
Cromi A, Bonzini M, Uccella S. et al. Provider contribution to an episiotomy risk model. J Matern Fetal Neonatal Med 2015; 28: 2201-2206 doi:10.3109/14767058.2014.982087.
Howden NLS, Weber AM, Meyn LA. Episiotomy use among residents and faculty compared with private practitioners. Obstet Gynecol 2004; 103: 114-118 doi:10.1097/01.AOG.0000103997.83468.70 

These two points should be addressed or commented on appropriately. I would also highlight in what way new evidence is added to the established evidence. 

I recommend a revision. 

With best regards

Author Response

Review 2

Dear Reviewer

Thank you very much for your thorough analysis of our article and for all valuable comments and tips.

The indicated remarks have been made in the manuscript. Details below:

  1. “So far, I do not understand how your study provides significant new information. Shmueli et al reported in 2017 in Shmueli A, Gabbay Benziv R, Hiersch L. et al. Episiotomy - risk factors and outcomes. J Matern Fetal Neonatal Med 2017; 30: 251-276 doi:10.3109/14767058.2016.1169527 on risk factors. 
    New additions were male sex and use of oxytocin. 

Thank you for your attention. The above publication concerns studies that were carried out in 2007-2014, while our study covers data from 2015-2020. In addition, our research updates data on risk factors. Currently, special attention is paid to the legitimacy of the use of medical interventions during childbirth, which is emphasized, among others, by the research of Çalik et al. (2018) [1]. The aim of our research was to search for risk factors for perineal incision in women giving birth in Poland. Based on the literature, we chose the risk factors analyzed so far, which we wanted to relate to our study group.

[1]Çalik, K.Y., Karabulutlu, Ö. & Yavuz, C. First do no harm - interventions during labor and maternal satisfaction: a descriptive cross-sectional study. BMC Pregnancy Childbirth 18, 415 (2018). https://doi.org/10.1186/s12884-018-2054-0

------------------------------------------------------------------------

  1. “Particularly with oxytocin as an influencing factor, I miss the discussion of causation. Does oxytocin also affect pelvic floor muscle tone, causing a delay in the child's egress?”

Thank you for your suggestion, we have expanded the discussion with the aspect indicated by the Reviewer

------------------------------------------------------------------------

  1. “I find the reference to a lower risk of episotomy in a birth center problematic without comment. The birth center is run by midwives who are fundamentally very critical of intervention in terms of episiotomy. Similarly, one could argue that midwives rarely perform apendectomies. 
    Here are two articles from a separate review on the topic: 
    Cromi A, Bonzini M, Uccella S. et al. Provider contribution to an episiotomy risk model. J Matern Fetal Neonatal Med 2015; 28: 2201-2206 doi:10.3109/14767058.2014.982087.
    Howden NLS, Weber AM, Meyn LA. Episiotomy use among residents and faculty compared with private practitioners. Obstet Gynecol 2004; 103: 114-118 doi:10.1097/01.AOG.0000103997.83468.70 

These two points should be addressed or commented on appropriately.

Thank you for your suggestion, the midwives aspect was extended in the discussion, we also took into account the publications indicated.

-------------------------------------------------------------------------------

  1. “I would also highlight in what way new evidence is added to the established evidence”. 

The presented work was created out of the need to search for scientific evidence based on practice. The results of our research and that of other authors constitute an important contribution to the continuing education of medical personnel (midwives and doctors) and the antenatal education of women. We have summarized this approach in the last paragraph of the Conclusions section. „Evidence-based preconceptive health education can help reduce unnecessary medical interventions and thus protect women’s reproductive health.”

Thanks again for all comments,

I hope that all the corrections made will be accepted.

Yours sincerely

Reviewer 3 Report

Thank you for the chance to review this article.

From my point of view the study is well organised and reported. The introduction is adequate. The methodology is well structured. The results are detailed and performed upon a logic plan and correlated with the study design. The discussion session is well documented and detailed. The conclusions are relevant.

I recommend the publication of the article in the present form.

Author Response

Dear Reviewer

Thank you very much on behalf of my team for the positive review of our article.

Yours sincerely
